# DO TISSUE SOURCE SITES LEAVE IDENTIFIABLE SIGNATURES IN WHOLE SLIDE IMAGES BEYOND STAINING?

**Piotr Keller, Muhammad Dawood & Fayyaz ul Amir Afsar Minhas**
Department of Computer Science
University of Warwick
Coventry
{Piotr.Keller,Muhammad.Dawood,Fayyaz.Minhas}@warwick.ac.uk

## ABSTRACT

*Why can deep learning predictors trained on Whole Slide Images fail to generalize?* It is a common theme in Computational Pathology to see a high performing model developed in a research setting experience a large drop in performance when it is deployed to a new clinical environment. One of the major reasons for this is the batch effect that is introduced during the creation of Whole Slide Images resulting in a domain shift. Computational Pathology pipelines try to reduce this effect via stain normalization techniques. However, in this paper, we provide empirical evidence that stain normalization methods do not result in any significant reduction of the batch effect. This is done via clustering analysis of the dataset as well as training weakly-supervised models to predict source sites. This study aims to open up avenues for further research for effective handling of batch effects for improving trustworthiness and generalization of predictive modelling in the Computational Pathology domain.

## 1 INTRODUCTION

Computational Pathology (CPath) is an emerging field which aims to leverage the ever increasing amount of health data to solve complex and clinically relevant problems through the application of machine learning (Abels et al., 2019). Areas of interest include, but are not limited to, predicting diagnostic abnormalities associated with cancer,nuclei instance segmentation and classification (Graham et al., 2021), cellular composition (Dawood et al., 2021), gene mutations and expression levels (Coudray et al., 2018; Dawood et al., 2022), as well as survival prediction of patients (Chen et al., 2022a; Mackenzie et al., 2022). In the recent years, there has been a growing popularity for the use of deep learning methods in CPath which utilise digitised slide tissue images also known as Whole Slide Images (WSIs) (Yao et al., 2020; Chen et al., 2022b; Lu et al., 2022). Such models have been very successful with many studies reporting a multitude of very high performance metrics on a wide range of datasets (Sokolova et al., 2006). Nevertheless, when some of these models are applied in a clinical setting they can fail to generalize (Foote et al., 2022).

In this work we argue that this is partially a consequence of reliance on stain normalization methods in the WSI pre-processing pipeline which are not able to truly remove the variability present in images sourced from different hospitals or laboratories. This variability occurs in the creation of these WSIs. As cells are transparent, it is necessary to stain tissue samples before they are digitised or observed under a microscope to effectively interpret them visually. The staining reagent and process as well as the scanner used can vary across source sites which often results in inconsistent staining characteristics across WSIs. This site-specific signature results in a batch effect which can be exploited by a deep learning model to produce inflated accuracy values but poor generalization in cases where the stain characteristics are different. Stain normalization approaches aim to remove this variability by reducing colour and intensity variations present in these images by normalizing them to a standard or base image. However, as stain normalization usually works in a low-dimensional space, we hypothesise that it fails to remove any higher order site-specific signatures which can still lead to exploitation of the batch effect and generalization failure under domain shift. This means

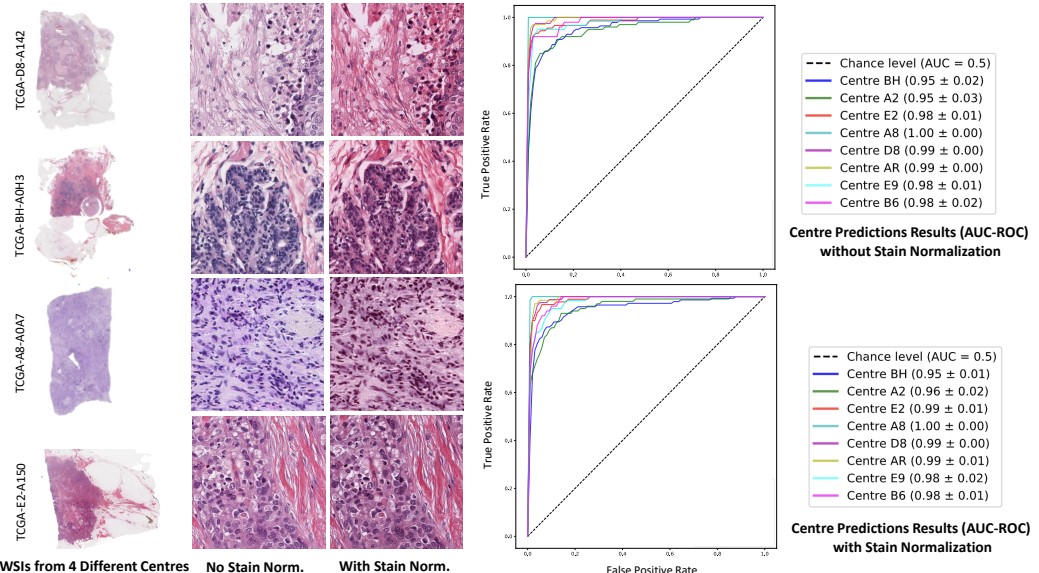

Figure 1: The left of the figure shows a visualisation of four WSIs originating from different source sites to highlight the staining variations found across source sites. For each WSI we show an example non-stain normalized patch and also a corresponding Macenko stain-normalized version to visualise how stain normalization appears to remove staining variation. The right of the figure, ROC cures are displayed for predicting tissue source site of WSIs using both non-stain normalized and stain-normalized images. The ROC curves are generated using an SVM that utilizes a pre-computed Maximum Mean Discrepancy kernel computed with ShuffleNet features (Keller et al., 2023b).

that stain normalized images look normalized to the human eye when in reality hidden factors such as those that result from different laboratory protocols are still present. These factors can skew the learning process acting as confounding variables which could lead to an overestimation of a model's true performance on a given task and subsequently be the cause of the poor generalisation in clinical deployment.

The technical contribution of this paper is the demonstration of empirical evidence of the presence of these hidden factors in a dataset regardless of what stain normalization technique is applied to it. This is achieved through a carefully designed experiment using different stain normalization schemes as well as two fundamentally different types of predictors. This aspect of the design of CPath pipelines is often ignored and, to the best of our knowledge, has not been previously explored with a carefully controlled experiment. The findings in this paper have significant bearing on improving trustworthiness and generalization of machine learning applications in the rapidly emerging area of CPath.

## 2 MATERIALS AND METHODS

### 2.1 EXPERIMENT DESIGN STRATEGY

To illustrate that stain normalization methods are unable to effectively remove center specific batch effects, we designed a simple experiment in which we predict the laboratory of origin (centre) of a WSI both before and after stain normalization. We first predict the centre of origin of a WSI by modelling this task as a weakly-supervised binary classification problem with the target label being the centre of origin. We then develop a separate weakly-supervised predictor to predict the centre of origin with stain normalized WSIs as input. The fundamental principle behind this experiment is that if stain normalization is an effective strategy to remove any identifiable signatures of the centre of origin or the underlying batch effect, we should get substantial decrease in accuracy of predicting the center after stain normalization. For this purpose, we use multi-centric breast cancer WSIs from the Cancer Genome Atlas (TCGA) (13 et al., 2012). To show that the results of our analysis are not

specific to a certain type of stain normalization, we utilize two different yet commonly used stain normalization schemes (Reinhard (Reinhard et al., 2001) and Macenko (Macenko et al., 2009)). In order to marginalize the effect of the choice of the weakly supervised method being used for the prediction of the centre of origin, we use two fundamentally different types of predictors (CLAM (Lu et al., 2021) and MMD-Kernels (Keller et al., 2023a)). Below each component of the experiment design is explained in further detail.

## 2.2 DATASET

1,113 publicly available WSIs of Formalin-Fixed paraffin-Embedded (FFPE) Hematoxylin and Eosin (H&E) stained tissue sections of 1084 breast carcinoma patients were collected from The Cancer Genome Atlas (TCGA-BRCA) (Hoadley et al., 2018; 13 et al., 2012). For some patients multiple WSIs were available and thus only the ones with best visual quality were used. Additionally, WSIs with missing baseline resolution information were ignored. After filtering 1,051 WSIs remain which are used for analysis. These WSIs were belonging to 49 sources sites.

## 2.3 PRE-PROCESSING OF WSIS

Quality of WSIs can be negatively affected by artefacts (tissue folds, pen-marking, etc) initiating from histology laboratories. To ensure that any models do not exploit these tissue artefacts the tissue regions of WSIs are segmented using a tissue segmentation method. The tissue segmentation means that only information tissue regions remain and artefacts are removed. Since, an entire WSI at full resolution can be very large ($100,000 \times 100,000$ pixels) and cannot be fitted into a GPU memory each WSI is tiled into patches of size $512 \times 512$ at a spatial resolution of 0.50 microns-per-pixel (MPP). Tiles that capture less than 40% of informative tissue area (mean pixel intensity greater than 200) are filtered out.

## 2.4 TISSUE STAINING AND STAIN NORMALIZATION METHODS

Histology images are acquired by staining a tissue specimen with a dye that shows variable affinities to different tissue components. In case of routine Hematoxylin and Eosin (H&E) staining, nuclei are stained with Hematoxylin and are highlighted in bluish color, while cytoplasm and extracellular matrix are stained with Eosin and can be seen in pinkish color (Fischer et al., 2008). However, due to variations in staining protocols, characteristics of the dye, duration for which the dyes are applied, tissue type and thickness, scanner characteristics and a number of other factors can impact the stain characteristics of the tissue resulting in center-specific confounding factors which are not at all related to any underlying pathology. These constitute a batch effect that can leave a centre-specific signature in the tissue image and affect the generalization performance of any machine learning method.

One way of addressing such variations is stain normalization using methods such as the ones proposed by Reinhard (Reinhard et al., 2001) and Macenko (Macenko et al., 2009). These stain normalization methods map the color style of source image to target images (Khan et al., 2014) while preserving cellular and morphometric information contained in the images.

## 2.5 SOURCE SITE PREDICTION

We hypothesise that majority of stain normalization methods try to make the images look similar but even after stain-normalization the histology laboratory from which tissue specimen is originating can still be predicted. More specifically, we argue that the use of stain normalization methods are not likely to make CPath algorithms generalize in case of domain-shift as these methods can not completely eliminate stain-specific information of the source site. To illustrate this, we used stain-normalized and non stain-normalized images and tried to predict the tissue source site as target variable. The hypothesis is that, if stain-normalization is removing the tissue site specific information then the tissue source site should be significantly less predictable from the stain-normalized images compared to non-stain-normalized images.

We demonstrated the predictability of tissue source site from stain-normalized and non-stain normalized images using a multiple instance learning method and also a kernel based method. As

a multiple instance learning method, we used Clustering-constrained Attention Multiple Instance Learning (CLAM) which is a weakly-supervised method that has shown promising performance in several computational pathology tasks (Lu et al., 2021). CLAM considers each WSI as a bag of patches and then used attention-based pooling function for obtaining slide-level representation from patch-level representation. As a second predictive model, we used a recently published support vector machine (SVM) based classification method that constructs a whole slide level kernel matrix using Maximum Mean Discrepancy (MMD) over ShuffleNet derived feature representations of patches in WSIs (Keller et al., 2023a). In recent work, this method has been shown to have strong predictive power for TP53 mutation prediction and survival analysis from WSIs. Note that the two methods have fundamentally different principles of operation so that any subsequent findings can be understood in a broad context independent of the specific nature of the predictive model being used. As it is not the goal of this work to present these specific predictors, the interested reader is referred to their original publications for further details.

We evaluated the performance of both these methods in predicting tissue source site using both stain-normalized and non-stain normalized data. The experiments were performed using stratified five fold cross validation. For each source site we train a separate model using one-vs-rest approach, in which all tissue images of patients originating from a given source site $L$ are labelled as 1, while the rest are labelled as 0. We then train the predictive model for predicting the source site of each WSI. In order to make meaningful comparisons, we restricted our analysis to prediction of 8 sources sites each of which has 50 or more images each.

The hyper-parameters were selected by utilising a validation set (30% of each train split fold). Average Area under the Receiver Operating Characteristic curve (AUC-ROC) across the 5 folds along with its standard deviation was used as the performance metric.

## 2.6 SIMILARITY KERNEL AND CLUSTERING ANALYSIS

In order to further understand the implication of stain normalization at a dataset level, we performed hierarchical clustering over the WSI MMD kernel matrix for the whole dataset. The matrix shows the degree of pairwise similarity between WSIs. We show the kernel matrices both before and after stain normalization together with clustering. If the stain normalization had been effective at removing any information about the center, we would expect that any clustering done after stain normalization will not be possible to group WSIs from the same center into the same cluster. This serves as an additional un-supervised analysis of whether clustering is able to remove center-specific information or not.

## 3 RESULTS AND DISCUSSION

### 3.1 EFFECT OF STAIN NORMALIZATION

Figure 1 shows the visual results of applying stain normalization to a patch belonging to 4 example WSIs each originating from a different centre. From the figure it can be clearly seen that patches belonging to different centers look the same after normalization hence to the human eye it would seem that we have removed the batch effect. However looking at the ROC curves we can see that both before and after stain normalization MMD kernels can near perfectly distinguish the WSI origin. This supports our hypothesis that stain-normalization methods are not removing the source site information. Even if after stain normalization the WSIs look the same the underlying footprint is still there. If stain normalization methods have truly removed source site information, then we will be seeing AUC-ROC of 0.5 (random) but this is not the case. From this analysis we can say that, the analyzed stain-normalization methods are less likely to make models robust against domain-shift.

### 3.2 PREDICTIVE POWER OVER ORIGINAL DATA

Tables 1-2 show the results of prediction of the source center from original WSIs, i.e., without any stain normalization using two different predictive pipelines (CLAM in Table-1 and MMD Kernel in Table-2). These results show that it is possible to predict the source of a given WSI with very high predictive power as measured using AUC-ROC for both methods. This shows that, as expected, there is a significant signature in a WSI specific to the laboratory of origin.

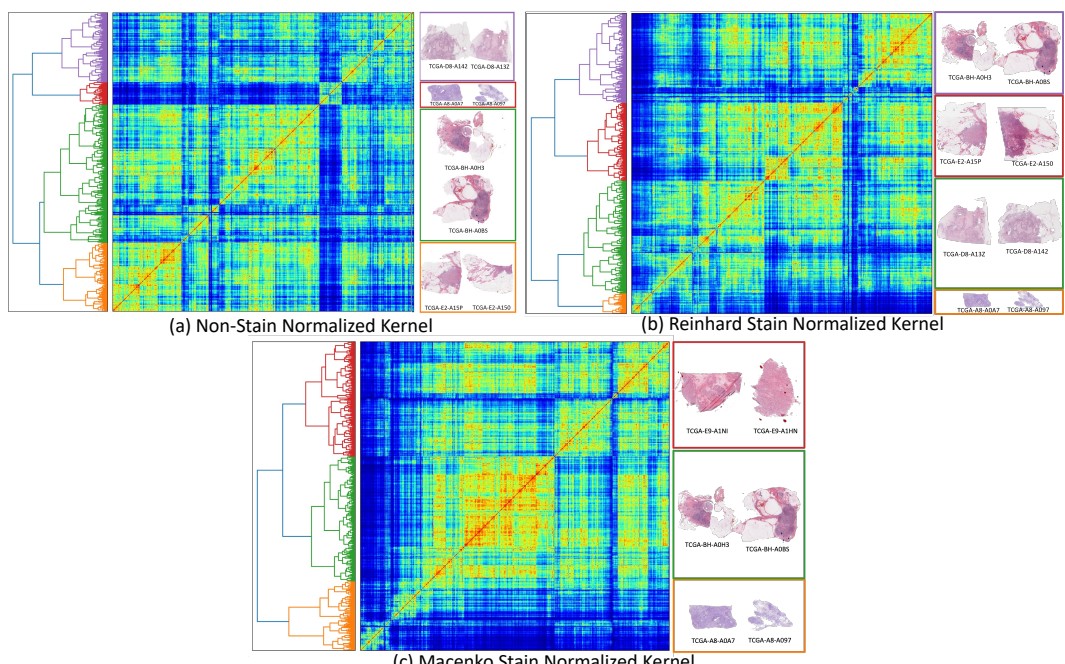

Figure 2: The visualisation of the kernels obtained by computing the MMD kernel on whole TCGA Breast cancer cohort. a) Show the pairwise similarity of WSI using non-stain normalized data, while b) and c) show the kernel matrices for Reinhard and Macenko stain normalized WSIs respectively. Each kernel also has an associated dendrogram as well as a visualisation of some of the patches from each of its clusters.

### 3.3 PREDICTIVE POWER OVER STAIN NORMALIZED DATA

Tables 1-2 show the results of prediction of the source center from stain normalized WSIs, i.e., with stain normalization using two different predictive pipelines (CLAM in Table-1 and MMD Kernel in Table-2) and two different stain normalization methods (Reinhard and Macenko stain normalization). These results show that it is possible to predict the source of a given WSI with very high predictive power as measured using AUC-ROC using both predictive pipelines even after stain normalization. There is effectively very little change in predictive power as a consequence of stain normalization. This shows that stain normalization alone is not able to remove the site-specific information contained in a WSI and the batch effect still exists even after stain adjustment.

### 3.4 CLUSTERING ANALYSIS

The hierarchically-clustered heatmaps along with their respective dendrograms for the kernels are shown in Figure 2. From this figure one can see that both non-normalized and stain normalized WSIs, have a large proportion of brightly coloured regions in their heatmaps indicating that there are many slides that share similar characteristics. The dataset has been split into 4 main clusters as can be seen on the dendrograms where slides within the same cluster seem to regularly originate from the same laboratory, for example the orange cluster contains many slides from laboratory E2 (Roswell Park). This indicates to us that some hidden site identification markers are likely to still be present even after normalization.

### 3.5 CODE AND DATA AVAILABILITY

The code and data used in this paper are publicly at :https://github.com/pkeller00/Src-Site-Pred.

Table 1: Comparison of performance of CLAM trained for source site prediction for various stain normalization protocols. Here + indicates WSIs that originated from the chosen site and − indicates WSIs from one of the remaining source sites.

| Source Site | (+,-) | Unnormalized AUC-ROC ± std | Reinhard AUC-ROC ± std | Macenko AUC-ROC ± std |
|---|---|---|---|---|
| University of Pittsburgh (BH) | (142,903) | 0.84 ± 0.04 | 0.82 ± 0.06 | 0.86 ± 0.03 |
| Walter Reed (A2) | (100,945) | 0.82 ± 0.10 | 0.73 ± 0.07 | 0.87 ± 0.07 |
| Roswell Park (E2) | (90,955) | 0.96 ± 0.01 | 0.92 ± 0.02 | 0.96 ± 0.01 |
| Indivumed (A8) | (74,971) | 1.00 ± 0.00 | 0.99 ± 0.00 | 1.00 ± 0.00 |
| Greater Poland Cancer Center (D8) | (78,967) | 0.97 ± 0.03 | 0.94 ± 0.04 | 0.97 ± 0.02 |
| Mayo (AR) | (69,976) | 0.98 ± 0.02 | 0.95 ± 0.04 | 0.97 ± 0.03 |
| Asterand (E9) | (62,983) | 0.98 ± 0.02 | 0.98 ± 0.01 | 0.96 ± 0.04 |
| Duke (B6) | (50,995) | 0.97 ± 0.03 | 0.92 ± 0.04 | 0.94 ± 0.06 |
| Average AUC-ROC | | 0.94 ± 0.08 | 0.94 ± 0.06 | 0.91 ± 0.09 |

Table 2: Comparison of performance of an SVM with a precomputed MMD kernel trained for source site prediction for various stain normalization protocols. Here + indicates WSIs that originated from the chosen site and − indicates WSIs from one of the remaining source sites.

| Source Site | (+,-) | Unnormalized AUC-ROC ± std | Reinhard AUC-ROC ± std | Macenko AUC-ROC ± std |
|---|---|---|---|---|
| University of Pittsburgh (BH) | (142,903) | 0.95 ± 0.02 | 0.93 ± 0.02 | 0.95 ± 0.01 |
| Walter Reed (A2) | (100,945) | 0.95± 0.03 | 0.88 ± 0.04 | 0.96 ± 0.02 |
| Roswell Park (E2) | (90,955) | 0.98 ± 0.01 | 0.98 ± 0.02 | 0.99 ± 0.01 |
| Indivumed (A8) | (74,971) | 1.0 ± 0.00 | 1.0 ± 0.00 | 1.0 ± 0.00 |
| Greater Poland Cancer Center (D8) | (78,967) | 0.99 ± 0.00 | 0.99 ± 0.01 | 0.99 ± 0.00 |
| Mayo (AR) | (69,976) | 0.99 ± 0.00 | 0.98 ± 0.01 | 0.99 ± 0.01 |
| Asterand (E9) | (62,983) | 0.98 ± 0.01 | 0.98 ± 0.01 | 0.98 ± 0.02 |
| Duke (B6) | (50,995) | 0.98 ± 0.02 | 0.97 ± 0.01 | 0.98 ± 0.01 |
| Average AUC-ROC | | 0.98 ± 0.02 | 0.96 ± 0.04 | 0.98 ± 0.02 |

## 4  CONCLUSIONS AND FUTURE WORK

We conclude that tissue source sites leave identifiable markers that can be picked by machine learning models. We show that this may be one of the reasons why many models often result in poor generalization when used outside a research setting thus we urge computational pathologists to keep this in mind when designing models and datasets. In the future we would like to verify our results on a larger database as well as explore what exactly are the most prominent factors that make a source site so easily distinguishable and how we can develop strategies to counter such confounding factors. On top of this we would like to see if there is a performance change when a model takes into account these source site signatures during training.

## ACKNOWLEDGEMENTS

FM acknowledges funding from EPSRC EP/W02909X/1 and PathLAKE consortium. MD and FM report research funding from GlaxoSmithKline outside the submitted work.

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
