# OpenReview forum: "Do Tissue Source Sites leave identifiable Signatures in Whole Slide Images beyond staining?"
_ICLR.cc/2023/Workshop/TML4H — ICLR 2023 Workshop TML4H Oral_

### Official Review · Reviewer_DwTK · 2023-02-24
**This paper provides empirical experiments on the domain problems of Computational Pathology**

**Rating:** 7
**Confidence:** 3

**Review:**

This is generally an interesting paper that proved the stain normalisation methods' less effectiveness on the batch effect and generalisation ability. Opened up the avenues for the researcher in this field.

Some impressive bullet points have been pointed out.
1. Stain normalisation methods cannot truly remove the variability/difference in images sourced from different hospitals or labs.
2. Stain normalisation cannon deal with the high-level order of site-specific signatures.

This paper looks more like an empirical study with comprehensively designed experiments. It does raise a new and usually ignored problem in the WSI field. Thus, I recommend accepting the publication.

---

### Official Review · Reviewer_GGfD · 2023-03-01
**The solved problem is interesting and provides new insight of low generalization ability in CPath.**

**Rating:** 6
**Confidence:** 3

**Review:**

Strength:
1) This paper is well-written and easy to understand.
2) It conducts extensive experiments to demonstrate that batch effect can't be effectively solved by stain normalization methods, which may be the reason for the low generalization ability in CPath. This problem is ignored by previous works for a long time.

Weakness:
1) The authors demonstrate that the images after stain normalization can be still successfully classified into their source sites. But it is not a direct clue to conclude that it is the root of low generalization ability in CPath. Why don't directly compare the generalization performance of models trained by original images and normalized ones on the images from the unseen source sites?
2) Does the images after data augmentation can be still correctly classified?

---

### Meta-Review · Area_Chair_ECoh · 2023-03-05

**Recommendation:** Accept (Poster)
**Confidence:** 5

**Metareview:**

As suggested by reviewers, this paper raises a new yet usually ignored problem in the field of WSI processing. While the methods and experiments can be further improved, reviewers suggested accepting this paper and allowing it to be reported in this workshop to inspire more studies.